# Sparsity-based compressive reservoir characterization and modeling by applying ILS-DLA sparse approximation with LARS on DisPat-generated MPS models using seismic, well log, and reservoir data

5 Mohammad Hosseini<sup>1,2</sup>, Mohammad Ali Riahi<sup>1</sup>

<sup>1</sup>Institute of Geophysics, University of Tehran, Tehran, 14155-6466, Iran <sup>2</sup>Geophysics Department, National Iranian South Oilfields Company, Ahwaz, 61735-1333, Iran *Correspondence to*: Mohammad Hosseini (hosseini.mhmd@yahoo.ca)

Abstract. In the earth sciences, there is only one single true reality for a property of any dimension whereas many realization models of the reality might exist. In other words, a set of interpreted multiplicities of an unknown property can be found but only one unique fact exists and the task is to return from the multiplicities to the uniqueness of the reality. Such an objective is mathematically provided by sparse approximation methods. The term 'approximation' indicate the sufficiency of an interpretation that is close enough to the true mode, i.e. reality. In geosciences, the multiplicities are provided by multiple-point statistical methods. Realistic modeling of the earth

- interior demands for more sophisticated geostatistical methods based on true available images, i.e. the training images. Among available MPS methods, the DisPat algorithm is a distance-based MPS method which generate appealing realizations for stationary and nonstationary training images by classifying the patterns based on distance functions using kernel methods. Advances in nonstationary image modeling is an advantage of the DisPat method. Realizations generated by the MPS methods form the training set for the sparse approximation. Sparse
- approximation is consisted of two steps, i.e. sparse coding and dictionary update, which are alternately used to optimize the trained dictionary. Model selection algorithms like LARS are used for sparse coding. LARS optimizes the regression model sequentially by choosing a proper number of variables and adding the best variable to the active set in each iteration. Out of numerous training dictionary methods given in the literature, the ILS-DLA is a variant of the MOD algorithm where the latter is inspired by the GLA and the whole trained dictionary is
- sequentially updated by alternating between sparse coding and dictionary training steps. The ILS-DLA is different from the MOD for addressing the internal structure of the dictionary by considering overlapping or nonoverlapping blocks and modifying the MOD algorithm according to the internal structure of the trained dictionary. The ILS-DLA is faster than the MOD in the sense that it inverts for smaller blocks constructing the trained dictionary rather than inverting for the entire block. The subject of this paper is an integration study between sparse
- approximations from image processing and compressed sensing, multiple-point statistics from the field of geostatisitcs, and the geophysical methods and reservoir engineering from the branch of petroleum science. This paper specifically emphasizes the utilization of image processing in solving reservoir complexities and enhancing reservoir models.

## 1 Introduction

In geoscience, there is only one single true model for a property of the earth but numerous models of the real earth property exists. This concept is referred to as uncertainty in the geoscience and reservoir modeling. In geoscience and reservoir modeling it is mostly acknowledged to manipulate the models and find the best one among a finite

set of possible models by performing further experiments and observing the results. It is neither guaranteed that the best model of a finite set of models is actually the true model nor that it is the best practically achievable model. Trivially, the more available models as the population samples, the more it is guaranteed that the selected best model is a closer translation of the reality, i.e. the true model.

# 1.1 Model multiplicities by the MPS methods

- In reservoir characterization and modeling, providing a large set of population samples is practically impossible unless introducing stochasticity in the models using the multiple-points statistics (MPS) methods. It is known that simple two-point geostatistics methods (Deutsch and Journel, 1998; Strebelle, 2000) fail to properly generate stochastic models especially for complex reservoirs, e.g. a reservoir which is deltaic in one part and fractured carbonate ramps in other parts. Delta properties are nonstationary and fracture network properties can also be
- nonstationary. Two-point geostatistics methods use variogram to model property variations between two points in the reservoir based on spatial distance. It will fail to properly model for farther distances if the changes in property is abrupt or nonstationary. To alleviate, different variogram models can be built for a neighborhood to capture property variations, but even this method does not completely capture the variations.
- As a more advanced variant, a location-independent model can be used as a source of information and statistical 20 changes for multiple-point statistics (MPS) algorithms to train and produce reliable reservoir static models. This location-independent model is called training image and its location-independence property will lead to producing various stochastic reservoir models (Haldorsen and Damsleth, 1990; Mariethoz and Caers, 2015). More realistic training images and more reliable stochastic models will be generated if wide and accurate prior information about the reservoir is available. Pattern-based methods like SimPat (Arpat, 2005), FilterSim (Zhang, 2006), Direct
- Sampling (Mariethoz and Renard, 2010), DisPat (Honarkhah, 2011), and WaveSim (Chatterjee et al., 2012) are generally best methods for MPS reservoir modeling. The SimPat algorithm is time-consuming and in the FilterSim, defining the filters adds complexity to the algorithm. The WaveSim algorithm is similar to the FilterSim and its complexity is in selecting an optimum scale for wavelet decomposition in the task of pattern classification. DisPat algorithm is known as the best of this kind for its ability to integrate data based on visual system of the human 30
- being and representing new algorithms for modeling images with nonstationary properties.

## 1.2 Sparse approximation as mathematical tools to sparsity-based compressed model

The MPS methods are used to generate the set of multiplicities based on one single training image which itself is not known as the true model. Therefore, the MPS generated realizations are all translation of an unknown true model. The task is to achieve one single model image as the representation of the true model from a large set MPS

realizations which are considered as the manipulations of the true model. The mathematical tools to perform such 35 a task is known as the sparse approximation from the field of image processing. The term approximation reminds the impossibility or unnecessity of obtaining the exact true model.

In a wider branch of image processing, the large set of stochastic reservoir MPS model images (generated by any of the MPS methods) could be used as a set of training images for dictionary training in the context of image compression (Aharon et al., 2006a and 2006b; Bryt and Elad, 2008; Cheng, 2015; Elad, 2010; Elad and Aharon, 2006; Horev et al, 2012; Khaninezhad and Jafarpour, 2013; Mairal et al., 2009 and 2008; Rubinstein et al., 2010a,

- 5 2013, 2010b and 2008; Skretting and Engan, 2011a, 2011b and 2010; Skretting and Husøy, 2003; Skretting et al., 1999; Starck et al., 2010). There are generally two types of dictionaries that can be employed to compress a set of model images: explicit and implicit dictionaries. The explicit dictionaries are out of the shelve dictionaries with fixed elements usually explicitly stated by analytical equations. The implicit dictionaries are different from the explicit ones in the sense that they are trained and adapted by the specific set of training images. Therefore, the
- 10 explicit dictionaries are general but less effective and the implicit dictionaries are trained on specific training images but more effective on that set or similar sets of training images.

The process of sparse approximation basically deals with implicit dictionaries which are trained and adapted on specific sets of training samples (images or signals). The sparse approximation is itself a sequential alternation between two steps: sparse coding and dictionary update. The sparse coding step deals with selecting the best vector

(or model) from a large set of vectors (or models) which minimizes an optimization problem under some sparsity constraint. In other words, in each iteration, the sparse coding methods like BP (Chen et al., 1999), MP (Mallat and Zhang, 1993), OMP (Rubinstein et al., 2010b; Tropp, 2004; Tropp and Gilbert, 2006), and ORMP (Gharavi-Alkhansari and Huang, 1998) and model selection algorithms like LARS (Efron et al., 2004; Tibshirani, 1996) and Lasso (Hastie et al., 2009) are employed using a fixed dictionary to obtain the sparse coefficients. The obtained sparse coefficients are alternately used in the dictionary update step.

Different methods of dictionary updating are presented in the literature, e.g. MOD (Cheng, 2015; Elad and Aharon, 2006), K-SVD (Aharon et al., 2006b; Bryt and Elad, 2008; Rubinstein et al., 2010b; Skretting and Engan, 2011a and 2011b), ODL (Mairal et al., 2009), ILS-DLA (Engan et al., 2007; Skretting and Engan, 2011b), and RLS-DLA (Skretting and Engan, 2011a and 2010). The MOD algorithm is inspired by the GLA to sequentially update the

- trained dictionary by alternating between the sparse coding and dictionary update steps inverting for the whole set of training data. The K-SVD algorithm is the generalization of the *k*-means clustering algorithm. In the K-SVD, the dictionary is updated sequentially by applying the singular value decomposition on each column of the trained dictionary in each iteration. The Online Dictionary Learning algorithm also follows the two step procedure and it is specifically designed to handle large scale data sets under the scope of online optimization algorithms. The ODL
- is fast for being based on stochastic approximation and updating the dictionary recursively by using the previous one as the warm restart. The ILS-DLA is a variant of the MOD algorithm and it is different from the MOD for addressing the internal structure of the dictionary by considering overlapping or non-overlapping blocks and modifying the MOD algorithm according to the internal structure of the trained dictionary. The ILS-DLA is faster than the MOD in the sense that it inverts for the smaller blocks which are constructing the trained dictionary rather
- than inverting for the whole block. The RLS-DLA exploits the same methodology as in the MOD algorithm where the RLS-DLA starts with an initial dictionary and at each step a new vector of data is introduced into the training set and the dictionary is updated accordingly. A forgetting factor is introduced into the algorithm to forget the early stages dictionaries and focus on the late stages dictionaries.

simpler two-point geostatistical methods.

## 1.3 The case study

In this paper, the sparsity-based compressive reservoir characterization and modeling workflow using the ILS-DLA training dictionary with LARS sparse coding methodologies is explained and applied on a gas injection case of an Iranian oilfield located southwest of Iran. The gas is injecting into the Asmari reservoir of this oilfield to maintain pressure and preserve the recovery factor. The Asmari reservoir in this oilfield is described as a very complicated reservoir because of the presence of interbeds of sandstones and carbonates, the possibility of a fracture network connecting the reservoir along the crest, and the presence of a deltaic depositional system in the west of the reservoir. Modeling such a complicated reservoir demands for new stochastic MPS methods rather than

- Seismic, well log and reservoir data are used to build semi-industrial reservoir model images and the DisPat MPS method is applied to manipulate the model images and generate a large set of stochastic reservoir model images for two cases of fracture and deltaic systems. The gas injection process is simulated on couple of 2D models extracted from the fractured part and the deltaic part of the reservoir. A set of stochastic realizations based on the training images from the fractured part were generated under the stationarity condition and for the deltaic part
- under the nonstationarity assumption. Each of these training sets were used to train dictionaries by the ILS-DLA method. The offline trained dictionary was then utilized in a sparsity-based image compression scheme using the LARS method to represent a single model, i.e. the compressed sparsity-based model. Proper experiments for comparison of goodness between the MPS generated stochastic realizations and the resultant compressed sparsity-based model image is to run the simulation for each model and compare the production and pressure profile with
- those of the true model. Experimental results show that the compressed sparsity-based model images, in almost every cases and testing all the methods, are superior to the majority of the MPS stochastic realizations, 89.58% of the experiments falling in the area of 90%-90% (upper 10%) superiority, and 95.83% in the area of 85%-85% (upper 15%). The results are even more encouraging considering the fact that the MPS realizations are quite stochastic and are not conditioned with hard or soft data, and that the interfering parameters involved in sparse approximation processes could be optimized to achieve much better results.

The subject of sparsity-based compressive reservoir characterization and modeling is an integration study which involves the geostatistics (for multiple-point statistics modeling), image processing (for dictionary training, sparse approximation, and image compression), seismic data interpretation and inversion (for providing AI along with saturation and porosity inverted 3D cubes, seismic spectral decomposition 3D cube, interpreted horizons), petrophysics (for providing well log data and interpretation), and reservoir simulation (for providing criteria to

quantify the goodness of model image).

## 1.4 Paper structure

In this paper, first the major blocks of the sparsity-based compressive workflow in reservoir characterization and modeling is presented. These main blocks are consisted of the DisPat multiple-point statistics method and the sparse approximation algorithm. The sparse approximation algorithm is consisted of sparse coding and dictionary training steps. For the sparse coding step, the LARS algorithm and its variants are introduced and for the dictionary update step, the ILS-DLA algorithm is represented. The application of the sparsity-based compressive workflow

on a case study is discussed afterwards and the results are presented. The paper is finalized by some concluding points.

## 2 DisPat MPS Algorithm

DisPat is a distance-based pattern-based multiple-point geostatistical method which was first introduced by
Honarkhah (2011). In this algorithm, alike the other pattern-based MPS methods, the training image is scanned by
the designed template and the patterns are extracted from the training image. The patterns are then classified based
on distance functions using the kernel methods. A distance function measures the distance between each pair of
patterns in the metric space. Any two close points in the metric space refer to two similar patterns from the pattern
database. The kernel *k*-means clustering algorithm is used to classify the patterns in the pattern database.

## 10 2.1 Pattern Simulation

Having the patterns classified, the sequential simulation is performed on the realization grid. The closest cluster to the data event is determined and one pattern from that cluster will be randomly chosen to be replaced on the node location. The distance function which is used to find the closest cluster to the data event has to account for the informed nodes, frozen nodes, and hard data nodes by specifying different weights to each type of the data. The

15 SimPat method uses Euclidean distance function to find the most similar pattern to the data event. Filtersim method uses the Manhattan distance function whereas DisPat algorithm uses two distance functions for two different purposes; the proximity distance transform is used to build the distance matrix and the Manhattan distance function is used to find the most similar pattern to the data event.

#### 2.2 Multigrid and Multiresolution

- To improve the long-range spatial characteristics of the realizations, DisPat utilizes the widely used multigrid approach and the newly invented multiresolution approach. In the multigrid approach, the grid size does not change but the template size changes from one level to another level. Sequential simulation starts from larger scale and the final realization at each level will be transferred to the lower level considered as informed nodes. As a result, large-scale properties are transferred from one grid level to the next finer grid level. In the multiresolution method,
- on the other hand, the template size is fixed and the grid size is changing from one level to the next. The simulation starts from the coarsest resolution and continues to the last finer resolution grid. The final realization at each level is informed to the next resolution level where a one-to-one correlation between two consequent grids is not held. Correspondingly, the realization and the training image are rescaled and interpolated to the next resolution level.

# 2.3 Hard Data Integration

The DisPat algorithm considers two hard data integration scopes for the two multigrid and multiresolution approaches. For the multigrid approach, the DisPat will strictly specify each hard data to the closest possible node and if the node has been introduced before, it will be specified to the next closest enclosing node. The hard data which are left out of this procedure will be rejected. In the multiresolution approach, the grids are different in size and the location of nodes changes from one level to the next level and a one-to-one correlation does not hold. The cubic interpolation is used to transfer the hard data from one level to the next level. For interpolation, either kriging

or inverse-distance weighting method is used. Hard data integration based on multiresolution approach will result in more appealing MPS realizations. The reason for such an improvement is that the multiresolution approach captures the statistical information of the training image better than the multigrid approach. Furthermore, the inverse-distance weighting will better represent the data structure in lower resolution grids.

# 5 2.4 Soft Data Integration

Hard data refers to well data and soft data refers to seismic data. In SimPat, along with the training image, a soft training image is considered and the soft pattern data base is sought and formed as it is done for the training image itself. A weight is given to the soft data to diminish or increase its role in building the final pattern based on the reliability wright given to the soft data. In Filtersim, Zhang (2006) and Wu (2007) proposed a new method for

- continuous images. They do the conventional unconditional simulation and find the most similar pattern for the node location. Simultaneously, the corresponding soft data event is found and pasted on the uninformed part of the data event and hence, the soft data is injected into the final simulation. For categorical images, the method in SimPat is integrated with the  $\tau$  model from the SNESim and a new data event is produced. The algorithm searches for the most similar prototype to the data event and the  $\tau$  model is used to combine this data event with the soft
- data event. Another search is done to find the closest prototype to this newest data event produced by the  $\tau$  model. In DisPat, the soft data integration is performed through distance calculations. The pattern databases are formed for the two sets of training image and the soft training image, patterns are clustered and the prototypes are specified. For each data event on a node location, the distance to each of the prototypes in either sets of the data (training image and the soft data) are calculated. These two distances are combined to form a new distance function in which
- the soft distance (i.e. the distance of the data event to the prototypes of the soft data) is weighted. This specified weight is itself a factor of the soft data reliability multiplied by the fraction defined as the ratio of the number of the informed nodes in a data event to the total number of nodes in that data event. It can be proven that this distance-based soft data integration is a special case of the  $\tau$  model.

# 2.5 Nonstationarity

- Training images could be stationary or nonstationary. In a stationary image the patterns are repeated and the statistical properties are steady, whereas in a nonstationary image, no prior information is expected for any location. In SimPat, a reservoir is subdivided into different regions upon the user's decision and based on the morphological characteristics of the reservoir. Each subregion is scaled or rotated such that the transition to the adjacent subregions are smooth and gradual. In Filtersim, additional morphological information is provided for grid nodes
- through the two other distinct grids conveying the scaling and rotation information. The drawback to these methods is the discontinuities which appear at the borders of these subregions where their corresponding training images are not coherent. Honarkhah (2011) has proposed three algorithms to generate nonstationary realizations needing only the training image as the prerequisite.

In the Spatial Similarity Method (SSM), the DisPat algorithm saves the patterns in the database along with the location of each pattern in a separate location database. This method is based on the fact that the patterns closer to the data event are more similar to it than the farther patterns. The SSM searches all over the grid for the most similar pattern. The most similar pattern to the data event is the one which minimizes a distance function and is

defined as combination of the weighted pattern distance with the weighted location distance. These weights are complementary and add up to 1. The neighborhood-radius method (NRM) is principally similar to the SSM method except that the algorithm searches within a definite circle of neighborhood and during the simulation, similar patterns inside this neighborhood will be searched for. The distance function will be the distance between the data

- event and the patterns; the location criteria is applied on the boundary within which the patterns are selected. This method is most suitable for semi-stationary training images where the circle of neighborhood controls the distance for which the statistical properties can be assumed stationary. The automatic segmentation method (ASM) is used for highly nonstationary images in which stationary regions cannot be defined for any border locations. As a result, such images must be automatically segmented into supposedly stationary regions. The approach to such a task is
- to extract several features of the property in the image and cluster the training image based on these extracted features by the *k*-mean clustering method. Gabor filter banks are used to extract image properties in few orientation angles at constant frequencies. The energy of each filtered images is then exposed by applying the sigmoid function and convolving with the Gaussian function as the smoothing filter. Additional Spatial component images are added to the set of features as input to the *k*-mean clustering algorithm. The *k*-means clustering algorithm will
- automatically segment the training image into k subregions based on the set of input features. The number of subregions, k, is defined by the user.

## **3** Sparse Approximation via Dictionary Learning

The exact way to represent the signal  $d \in \mathbb{R}^n$  as a linear combination of the columns of matrix  $G \in \mathbb{R}^{n \times k}$  is given by Gm = d, where  $m \in \mathbb{R}^k$ . The sparsest solution to this problem is expressed as  $\min_m ||m||_0$  for which the  $||.||_0$ 

counts the number of non-zero entries in the solution. The most desired solution to this system of equations is the one with the fewest number of nonzero coefficients.

In an image processing problem, matrix  $\boldsymbol{G}$  is called dictionary and its columns are called the atoms. These atoms are the building blocks of the dictionary and are normalized. Vector d can be represented as a linear combination of the dictionary atoms,  $\{G_i\}_{i=1}^k$ . Curvelets, contourlets, wedgelets, bandlets, steerable wavelets, and short-time

Fourier transforms are examples of the dictionary matrices. This representation is either exact or approximate and the exact representation of the vector *d* is expressed as

$$\min_{m} \|m\|_{0} \quad subject \ to \quad Gm = d \tag{1}$$

The solution to the exact problem is  $Q(\mathbf{G}^{-1}d)$  and this is only achievable if  $\mathbf{G}$  is invertible. In real problems, the exact solution is usually unachievable or it is not favorable and the approximate solution suffices. Obtaining the approximate solution to  $\mathbf{Gm} \approx d$  which bears the criterion of being the sparsest, is called the sparse approximation.

approximate solution to  $Gm \approx d$  which bears the criterion of being the sparsest, is called the sparse approximation The formulation for sparse approximation is represented as

$$\min_{m} \|m\|_{0} \quad subject \ to \quad \|Gm - d\|_{2} \le \epsilon \tag{2}$$

In this case, matrix G is an overcomplete dictionary. The whole process is called compression technique. In the well-known cases of transform coding, the DCT or wavelet dictionaries are used to compress the data. The compression process finds and captures the redundancies in the data. For a set of N signals, if the columns of G ∈ R<sup>n×k</sup> are normalized and m ∈ R<sup>k×N</sup> (n < k ≪ N) is sparse enough with s<sub>0</sub> non-zeros in each column, the solution

to the factorization Gm = d is guaranteed, it is unique, and it is the sparsest solution and can be achieved by the pursuit methods. Here,  $d \in \mathbb{R}^{n \times N}$  ( $n \ll N$ ) is the set of signals containing N signals as its columns and m is the compressed solution and it certainly has lower entropy than the signal d (Aharon et al., 2006a and 2006b; Horev et al, 2012; Rubinstein et al., 2010b).

Sparse transform has wide applications in compression, feature extraction, regularization in inverse problems, denoising, dynamic range compression in images, separation of texture and cartoon content in images, inpainting, facial imagery and more.

Optimal dictionaries are considered as flexible, simple, efficient for which the objective functions are well-defined. It is appropriate to have a structured dictionary rather than a free one. The proper dictionary is either drawn from

- a prespecified set of linear transforms or it is adapted to a set of training signals. The prespecified dictionaries are simple and fast and can be easily pseudoinversed if tight frames are used. These predetermined dictionaries are also called analytic dictionaries because the algorithm to derive the coefficients is known and analytically stated. For the same reason, they are called implicit dictionaries because the coefficients are implicit analytical statements instead of explicitly stated numbers. These dictionaries are highly structured, practically fast and efficient but they
- lack adaptability and they are only used for general-purpose image compression tasks.

Alternatively, the trained content-specific dictionaries outperform the prespecified ones. One drawback to the content-specific dictionaries is its computational time. Whereas this drawback is downplayed nowadays by the ever-growing computational capabilities, but still their drawback of loss of generality remains as these content-specific dictionaries are optimized for specific classes of images. These learning-based explicit dictionaries are

- adaptable but costly and inefficient. The coefficients of the learning-based dictionaries are explicitly computed by using machine learning algorithms which train the dictionary on a set of examples. The explicit dictionaries are tuned much finer than the implicit dictionaries and they perform significantly better. The size of the trained dictionary and the processed signal are limited by the complexity constraints.
- As a remedy, an input-adaptive approach can be designed to restore generality while preserving adaptivity. A novel dictionary, G, with a parametric structure is defined as the product of a fixed non-adaptive base dictionary,  $\phi$ , and a sparse atom representation matrix, S, i.e.  $G = \phi S$ . The choice of  $\phi$  imposes a structure on the process of dictionary training which acts as regularizer and reduces the overfitting and instability in the presence of noise. The universal base dictionary consists of a fixed set of fundamental signals from which all observable dictionary atoms are formed. The generic matrix  $\phi$  can have any number of atoms but they are assumed to span the signal
- space. The matrix S provides efficient forward and adjoint operators which bridges the gap between implicit and explicit dictionaries (Horev et al, 2012).

Adaptive methods prefer explicit dictionary representation over the structured ones. In fact, the atoms in G is a sparse combination of the atoms in  $\phi$ . In a sense, the dictionary G can be viewed as an extension to existing dictionaries (matrix  $\phi$ ) adding them a new layer of adaptivity. The adaptive structure (G) is significantly more efficient than the generic dictionary ( $\phi$ ) depending on the  $\phi$ -choice. It is evidently more compact to store and transmit. Parametric dictionaries bridges the gap between complexity and adaptivity for gaining a higher degree of freedom in training but sacrificing regularity and efficiency of the result. As a result, the input-adaptive approach

compressions are applicable to a wide range of images (Horev et al, 2012) and it provides a simple, flexible,

adaptive and efficient dictionary representation for its low complexity, compact representation, stability under noise, and reduced overfitting.

The main difficulty of the sparse representation is the problem of dealing with l<sup>0</sup> norm. The basis pursuit (Chen et al., 1999) turns this problem into an l<sup>1</sup> norm which is an optimization problem, i.e. Eq. (1) and Eq. (2) are convexicated by replacing l<sup>0</sup> semi-norm with an l<sup>1</sup> norm. Therefore, the goal of the basis pursuit is to find a solution to the sparse representation problem for which the number of non-zero coefficients is minimized. On the other hand, Lasso (Tibshirani, 1996) does not minimize the number of non-zero coefficients but it nulls the coefficients by changing the non-zero coefficients until the number of non-zero coefficients reaches a definite threshold. Lasso is in fact a sparsity-constrained sparse coding problem. Lasso is a modification of the Least Angle
Regression and Shrinkage (LARS) algorithm and both are described under the scope of model selection algorithms.

# 3.1 Model Selection Algorithms

Subset Selection, Forward Selection, Forward Stagewise Linear Regression, Backward Elimination, Lasso, and LARS are different types of model selection algorithms used to select a parsimonious set of covariates among a larger set of them. Subset Selection is a discrete process which produce interpretable models by retaining a subset

15 of predictors and discarding the rest. Shrinkage methods like ridge regression, Lasso, and LARS are more continuous than the subset selection method and are less susceptible to high variance. The followings are selected from (Efron et al., 2004; Hastie et al., 2009).

#### 3.1.1 Ridge Regression

The ridge coefficients are obtained by minimizing the residual sum of squares which are also penalized by 20 imposing a constraint on their size

$$\widehat{m}^{ridge} = \underset{m}{\operatorname{argmin}} \left\{ \sum_{i=1}^{n} \left( d_i - m_0 - \sum_{j=1}^{k} g_{ij} m_j \right)^2 + \lambda \sum_{j=1}^{k} m_j^2 \right\}$$
(3)

where  $\lambda$  is a shrinkage controlling complexity factor and its larger values leads to larger shrinkage. The following is an equivalent way to write the ridge problem and explicitly state the size constraint

$$\widehat{n}^{ridge} = \underset{m}{\operatorname{argmin}} \sum_{i=1}^{n} \left( d_i - m_0 - \sum_{j=1}^{k} g_{ij} m_j \right)^2 \quad subject \ to \quad \sum_{j=1}^{k} m_j^2 \le t$$

$$\tag{4}$$

25 The inputs to the ridge regression should be centered and standardized before solving for the coefficients. The ridge problem can be rewritten in matrix form as

$$RSS(\lambda) = (d - Gm)^T (d - Gm) + \lambda m^T m$$
<sup>(5)</sup>

And the solution will be

$$\widehat{m}^{ridge} = (\boldsymbol{G}^T \boldsymbol{G} + \lambda \boldsymbol{I})^{-1} \boldsymbol{G}^T d \tag{6}$$

30 This formula indicates that the ridge estimates are just the ordinary least squares solution scaled by a factor of  $(1 + \lambda)$ . Writing the SVD decomposition of the  $n \times k$  matrix **G** as  $U\Sigma V^T$ , the least squares fitted vector can be written as

$$\boldsymbol{G}\widehat{\boldsymbol{m}}^{ls} = \boldsymbol{G}(\boldsymbol{G}^T\boldsymbol{G})^{-1}\boldsymbol{G}^T\boldsymbol{d} = \boldsymbol{G}\boldsymbol{G}^T\boldsymbol{d} \tag{7}$$

and the ridge solution is equivalently