# Peer review of "Sparsity-based compressive reservoir characterization and modeling by applying ILS-DLA sparse approximation with LARS on DisPat-generated MPS models using seismic, well log, and reservoir data"

_Nonlinear Processes in Geophysics, 2016_

## Referee Comment (RC1) · Anonymous Referee #1 · 26 Oct 2016

Review of discussion paper, version Sep. 12, 2016, (downloaded Oct. 5)
**Sparsity-based compressive reservoir characterization and modeling by applying ILS-DLA sparse approximation with LARS on DisPat-generated MPS models using seismic, well log, and reservoir data**
Authors: *Mohammad Hosseini and Mohammad Ali Riahi*

Review done Oct. 26, 2016.

This reviewer works in the image and signal processing area and has no particular knowledge of geophysics or geoscience. Thus, this review will only consider the general appearance of the paper and the parts of the paper related to image and signal processing, especially sparse coding and dictionary learning.

The paper, in its current state, is unclear in most parts. There are several errors and concepts that are not correctly used or explained. The idea of the paper, to use sparse approximation on reservoir models, may be a good idea for further research and eventually one or several papers. This reviewer thinks that a major revision is needed before this paper is published. Perhaps it is better to start a new paper.

**The title** is too long and contains several abbreviations that one should not expect the reader to be familiar to. I think a better title could be something like: *Sparsity-based compressive reservoir characterization and modeling from seismic, well log, and reservoir data.*

I also think that **the abstract** is too long. You try to explain the methods here, and this is usually not a good idea as there are not enough space. Abbreviations should be written out first time they are used, one way to do this is, line 14 page 1, "... provided by multiple-point statistical methods (MPS).". Sometimes in the paper this convention is used, page 6 line 34, page 7 line 2 and 7. The part from end of line 19 to the middle of line 29 deals with sparse coding and dictionary learning. Here, you try to explain the two steps of dictionary learning both on line 20 and on line 25 and in between topics more or less related are mentioned, this gives a messy impression. Also, you write *sparse approximation* when it should be *dictionary learning*, this error is repeated several times in the paper and reveals that the concepts are not correctly understood. Finally, the relationship between MOD, ILS-DLA (and RLS-DLA) seems to be wrongly explained: ILS-DLA is not faster than MOD.

Before I continue, I will give some opinions on notation and explain some concepts:
It may be helpful for the authors to use a notation more in line with the notation common in image and signal processing, even though the notation is not well established and several variants are used. Each paper should define all used symbols. It is common to denote the dictionary as $D$ (or $C$ for codebook

or $F$ for frame) and atom $i$ as $d_i$, the signal as $x$ (or $y$ or $s$), the coefficients could be denoted in any way ($w$, $\alpha$, $y$, $x$, $\gamma$, $a$, ...), but not $m$ as the characters from $i$ to $n$ are usually used to represent integers. The approximation error can be denoted as $r$, $e$ or $v$. Using the notation below the fact that the signal is equal to its approximation plus the error can be written as $x = Dw + r$.

A *dictionary* (or codebook) is a generalization of a basis and a frame. It is simply a collection of elements, denoted as *atoms*, in a (Hilbert) space. The atoms do not need to span the space, as they should do for a basis and a frame. The space considered is often $\mathbb{R}^N$ and it is common to restrict the atoms to be of unit norm (2-norm). *Sparse Coding* (model selection or vector selection) is to find the coefficients that gives a sparse representation (if exact, no error) or a sparse approximation (if a small error is allowed). *Dictionary learning* or dictionary training is to learn a dictionary based on a set of training data. This process is often done iteratively starting with an initial dictionary and then iteratively repeat two main steps: 1) Keep the dictionary fixed and find the sparse coefficients (Sparse Coding). 2) Fix the coefficients and find the dictionary.

**The introduction** should present the motivation and give the context of the paper. It can also give a brief overview of methods and algorithms presented elsewhere and that the current paper builds on, this may be done in separate sections. In this paper the introduction is too long, as it ends on page 16. Perhaps it also includes some of the papers new ideas, but this is impossible for a reader to detect. Section 1.3 seems to describe the experiment done and presents some results, these parts should be clearly separated and moved towards the end of the paper.

I will not comment on the (introduction) sections that describes MPS algorithms (section 2) or sparse coding (section 3.1) but only comment what the authors call the ILS-DLA algorithm (section 3.2). The beginning of section 3 is only confusing; the concepts are not correctly used and it is difficult to see the relevance of most paragraphs (ex. page 8 line 24 to page 9 line 2). For a brief overview of some concepts you may look at the introduction of reference [Skretting K., Engan K., 2011b], otherwise this reference is not relevant for this paper. For a more comprehensive textbook, I recommend reference [Elad M., 2010.].

In section 3.2 the paper tries to give a summary of reference [Engan K., Skretting K. and Husy J. H., 2007]. It is not successful. As noted in the abstract the relationship between MOD, ILS-DLA (and RLS-DLA) is not understood. ILS-DLA is the name used to denote all (iterative) dictionary learning algorithms where the dictionary update step is done by minimizing the sum of square errors, i.e. finding the least squares (LS) solution. MOD was the first algorithm introduced that uses the LS approach. MOD **is** the block based variant of ILS-DLA, it is wrongly described in section 3.2.1. The matrix to be

inverted in Eq. (16) page 13 is large and not invertible, as $\mathbf{m}_{aug}^{(i)}$ is a $kL \times 1$ column vector in Eq. (12) on page 12. The correct way would be to stack the coefficient vectors beside each other to form a $k \times L$ matrix. There are also overlapping and constrained variants included in ILS-DLA, but I can not see any reason to include a description of these in sections 3.2.2 and 3.2.3. It is probably the block based variant that is used in the experiments, but this is not clear from the text.

After the introduction, I would expect to find a section describing **the method** that the paper presents. This should give a detailed presentation of the new research done and clearly present the contributions of the paper. The paper does not have any such section. Then I would expect to find a description of **the experiments** done to confirm the proposed method. In the end, **the results** should be presented followed by a discussion and a brief conclusion. The paragraphs belonging to the parts mentioned above are mixed together in sections 4 to 6 (and 1.3) in a way that makes it difficult to sort it out.

Finally, the reference list seems to be a randomly selected list of papers, many of these are excellent papers but I fail to see why some of them are included here, ex. [Skretting K., Husøy J.H., 2003], [Tropp J.A., Gilbert A.C., 2007], [Vidal R., Ma Y., and Sastry S., 2005] and several more.

---

## Referee Comment (RC2) · Anonymous Referee #2 · 31 Oct 2016

The general comments are:

The paper "Sparsity-based compressive reservoir characterization and modelling by applying ILS-DLA sparse approximation with LARS on DisPat-generated MPS models using seismic, well log, and reservoir data" include a workflow that is integrating multi-point statistics and some type of classification for reservoir characterization and

modelling. The workflow applied on one producing field with some hard and soft data. The results verified with different measures including subsequent drilling outcome. The topic is certainly the subject of international research and the access to the dataset that the authors had made it possible to carry out the study and test the integrated multi-point statistics and a type of machine learning workflow. The text is fluent and easy to read and understand for large part of the paper.

However, there are some issues with the structure and organization of the paper. These issues are: The standard structure of a paper including introduction, method-ology, result, discussion, and conclusion is not consistent and clear enough. Below is more detail for every section: In the introduction, we expect to see the state of the art with references. This is done but unnecessary numbering of the introduction made it very long. A brief of the method applied must be given which is not clear in the in-troduction. A simple guide can be found in: J. F. Claerbout, 1991, "A scrutiny of the introduction," The Leading Edge, 10, 39. There is a lot about the case study in the introduction which must be reduced significantly. So make introduction shorter, focus on state of the art for the main elements of the proposed workflow.

Methodology:

This needs significant change. Sections 2, and 3 to be included under methodology and reduced considerably. Since the main message of the paper is the workflow and not the development of the individual methods such as DisPat MPS algorithm, the mathematical formulations have to be reduced and only leave the main one which represent the approaches. This paper doesn't need to illustrate how the mathematical formulation for the used approaches developed.

In sections 2, and 3 it is unclear at the end which approaches has been selected by the authors. Methodology part must include clear sentences on what method has been used by the authors at the end of each part.

Methodology part also lacks proper explanation for soft and hard data preparation. This

is also part of the methodology. Approaches used for deterministic seismic inversion, spectral decomposition, and property classifications (e.g. porosity) are to be described briefly under methodology. So the methodology section has to be revised and new one includes: 1-MPS 2-Model selection 3-Soft and hard data preparation.

Illustrations:

Figures need to be improved to support the conclusions given in the paper. Figures lack scale, orientation, and in some cases, proper annotations. More comments on individual figures are given in the pdf file. On figure showing some representative well logs is needed in order to confirm the interpretation of delta type reservoir claimed in the paper. It means that wells that show the fluvial facies that has been interpreted on inverted and decomposed seismic data.

The discussion part of the paper has a lot of overlap with the results.

All in all the manuscript has certainly the good material to be published and it is not a difficult task to include the comments above and those mentioned in the pdf file in order to convey the main message of the paper.